# Structures of the scanning and engaged states of the mammalian SRP-ribosome complex

Rebecca M Voorhees, Ramanujan S Hegde*

MRC Laboratory of Molecular Biology, Cambridge, United Kingdom

**Abstract** The universally conserved signal recognition particle (SRP) is essential for the biogenesis of most integral membrane proteins. SRP scans the nascent chains of translating ribosomes, preferentially engaging those with hydrophobic targeting signals, and delivers these ribosome-nascent chain complexes to the membrane. Here, we present structures of native mammalian SRP-ribosome complexes in the scanning and engaged states. These structures reveal the near-identical SRP architecture of these two states, show many of the SRP-ribosome interactions at atomic resolution, and suggest how the polypeptide-binding M domain selectively engages hydrophobic signals. The scanning M domain, pre-positioned at the ribosomal exit tunnel, is auto-inhibited by a C-terminal amphipathic helix occluding its hydrophobic binding groove. Upon engagement, the hydrophobic targeting signal displaces this amphipathic helix, which then acts as a protective lid over the signal. Biochemical experiments suggest how scanning and engagement are coordinated with translation elongation to minimize exposure of hydrophobic signals during membrane targeting.

## Introduction

Roughly 20% of eukaryotic genes encode membrane proteins that must be inserted into the endoplasmic reticulum (ER) early in their biogenesis (*Krogh et al., 2001*; *Shao and Hegde, 2011b*). The majority of these proteins are recognized co-translationally at the ribosome by the signal recognition particle (SRP) and targeted via the SRP receptor to the Sec61 translocon for insertion (*Halic and Beckmann, 2005*; *Shan and Walter, 2005*). SRP has several critical functions in coordinating this targeting reaction. First, SRP must selectively recognize its cargo via a hydrophobic targeting signal. Second, it must shield this hydrophobic domain to preclude inappropriate interactions. Third, cargo-loaded SRP must interact with its receptor and release the cargo to the translocon. All of these events must occur before substantial synthesis beyond the targeting signal precludes successful engagement of the translocon (*Siegel and Walter, 1988*).

In eukaryotes, SRP is composed of six protein subunits assembled on a ~300 nucleotide RNA (*Keenan et al., 2001*). The elongated assembly is operationally divided into an S domain and an Alu domain (*Siegel and Walter, 1986*). The S domain is involved in recognizing the targeting signal and interacting with the SRP receptor. Its functionally essential SRP54 subunit contains three modules: an N domain that interacts with the ribosome near its exit tunnel (*Halic et al., 2004*, *2006*; *Schaffitzel et al., 2006*), an M domain responsible for signal recognition (*Zopf et al., 1990*; *Halic et al., 2006*; *Janda et al., 2010*), and a G domain that (together with the N domain) coordinates receptor targeting (*Miller et al., 1993*; *Egea et al., 2004*; *Focia et al., 2004*). The Alu domain interacts near the GTPase center at the interface between the two ribosomal subunits (*Halic et al., 2004*). This interaction is thought to compete with translation factors to slow nascent polypeptide elongation and provide additional time for successful targeting. Many prokaryotes lack an Alu domain and instead contain a simplified SRP composed solely of an SRP54 homolog and a short ~110 nucleotide RNA.

*For correspondence: rhegde@ mrc-lmb.cam.ac.uk

**eLife digest** Proteins are long chain-like molecules built from smaller building blocks, called amino acids, by a large molecular machine known as a ribosome. Although all proteins are assembled inside cells, some of them must be delivered to the outside or inserted into cell membranes. It is important to understand how this selective delivery system works because secreted proteins (i.e., those delivered outside) and membrane-embedded proteins are essential for cells to communicate with their surroundings. Proteins destined for secretion or membrane insertion contain characteristic stretches of amino acids that act as a targeting signal for delivery to the membrane. These targeting signals are recognized by the 'signal recognition particle' (or SRP for short), a large complex found in all living organisms. The SRP has the task of finding ribosomes that are assembling proteins with a targeting signal, and then taking them to the membrane. The protein being assembled can then either cross the membrane for secretion by the cell, or get embedded within the membrane.

So, how can the SRP scan the broad range of proteins that are made by the ribosome and engage with only those containing targeting signals? Voorhees and Hegde investigated this question by analyzing SRPs bound to ribosomes that were at different stages of building a membrane protein. The experiment was devised so that SRP would be in two different states: in the first state, the SRP was scanning for its targeting signal and, in the second, it was engaged with the targeting signal. Voorhees and Hegde took many thousands of pictures of these samples using a technique called cryo-electron microscopy, and reconstructed the three-dimensional structures of both states. This revealed fine details of how SRP positions itself immediately next to the part of the ribosome where newly formed protein chains emerge. From here, the SRP scans the protein until the targeting signal emerges and then it engages with the protein.

Engaging the targeting signal just as it emerges from the ribosome is probably important because targeting signals tend to aggregate if they are exposed to the contents of a cell. The new structures show how SRP cradles the targeting signal inside a binding groove and covers it with a protective lid to minimize its risk of aggregation. The next challenges are to figure out how SRP chooses which ribosomes to scan, and how it releases the targeting signal when it has delivered it to the membrane.

Despite extensive biochemical and structural studies, several basic aspects of SRP biology remain unresolved. First, the interactions between SRP and the ribosome are understood only in general terms, with regions of close proximity deduced from moderate resolution cryo-EM structures (*Halic et al., 2004*, *2006*; *Schaffitzel et al., 2006*). The specific residues and chemical details of the binding surfaces are not known. Furthermore, the eukaryotic SRP-ribosome interaction has only been visualized for mammalian SRP bound to the plant ribosome (*Halic et al., 2004*, *2006*). Although functional for translocation, this heterologous complex strongly arrests translation (*Walter and Blobel, 1981*). By contrast, endogenous SRP in a homologous mammalian system has a remarkably subtle effect on translation elongation (*Wolin and Walter, 1989*). Hence, the nature of the Alu domain-ribosome interaction and its relationship to translation factor binding in a native homologous system is unknown.

Second, the molecular events leading to signal recognition by the SRP M-domain are also incompletely resolved. Structural analyses of the M domain show that it contains a large hydrophobic groove (*Keenan et al., 1998*) capable of housing a hydrophobic signal (*Janda et al., 2010*). While this has convincingly established the binding site, the configuration of the pre-engaged state of the M domain and how it transitions to the engaged state are less clear. An exposed hydrophobic groove, as seen in isolated M domain crystal structures (e.g., *Keenan et al., 1998*), would seem energetically unfavorable in an aqueous environment and may exhibit promiscuous interactions. Indeed, other factors that bind hydrophobic clients contain hydrophobic grooves that are occluded in the unengaged state (*Pellecchia et al., 2000*; *Mateja et al., 2009*). Thus, it is not known how SRP's substrate binding groove is kept shielded until it encounters its hydrophobic clients specifically at the ribosomal exit tunnel.

Third, the timing of SRP recruitment to the ribosome relative to its engagement of the targeting signal remains a source of considerable debate. The simplest and earliest idea was that SRP has high affinity and selectivity for a ribosome exposing a hydrophobic signal (*Walter et al., 1981*).

However, such a model would not easily explain why far more abundant hydrophobic binding proteins do not outcompete SRP, why SRP cannot bind those same sequences post-translationally, and how SRP can rapidly find these transient and rare species to prevent aggregation. Thus, it was suggested that SRP may repeatedly scan all translating ribosomes until the appropriate cargo is encountered (*Ogg and Walter, 1995*).

Studies in the simplified bacterial system have extensively debated both the basic premise and mechanistic details of this scanning model (*Bornemann et al., 2008*; *Zhang et al., 2010*; *Holtkamp et al., 2012*; *Zhang and Shan, 2012*; *Saraogi et al., 2014*; *Noriega et al., 2014a*, *2014b*). The extent to which these prokaryotic studies apply to eukaryotic SRP is unclear and has been minimally studied. Not only is its ribosome interaction more complex, but the Alu domain merits consideration of how SRP recruitment is integrated into the translation cycle. Furthermore, *Escherichia coli* SRP seems to scan any translating ribosome (*Bornemann et al., 2008*; *Holtkamp et al., 2012*), while eukaryotic SRP may display a further preference for ribosomes containing a hydrophobic signal inside the exit tunnel (*Flanagan et al., 2003*; *Berndt et al., 2009*; *Mariappan et al., 2010*; *Zhang et al., 2012*). At present, the existence and properties of this scanning mode remain a point of uncertainty, but is crucial for understanding how hydrophobic signals are promptly found by SRP before their cytosolic exposure leads to off-pathway fates.

Here we demonstrate that endogenous SRP is efficiently recruited to mammalian ribosomes containing a transmembrane domain (TMD) inside the ribosomal exit tunnel. The cryo-EM reconstruction of this native complex provided the first structure of SRP bound to the ribosome in a 'scanning' mode. A parallel structure of SRP bound after the TMD has emerged from the ribosome revealed the conformational changes that accompany TMD engagement by SRP. The structures show how the hydrophobic cavity of the SRP M domain is occluded by an amphipathic C-terminal 'placeholder' helix until its displacement by a *bona fide* TMD. Competition assays between SRP and a translational GTPase provide insight into how scanning and engagement are integrated during polypeptide elongation to maximize targeting efficiency while limiting TMD exposure to the cytosol. These findings have functional implications for the mechanism of membrane protein recognition and targeting.

## Results

### Biochemical analysis of SRP complexes in scanning and engaged states

To examine the timing and specificity of SRP recruitment, a variety of stalled ribosome-nascent chain complexes (RNCs) were prepared by in vitro translation of truncated mRNAs in rabbit reticulocyte lysate, affinity purified via the nascent chain, and analyzed biochemically. Constructs and truncation points were designed to position a 20 residue TMD either within the exit tunnel or just emerged from the ribosome (*Figure 1A*). These two states were chosen to represent moments when SRP would be either scanning in anticipation of, or successfully engaged with, the TMD. Specificity was controlled by using a mutant (hereafter termed 3R) in which three hydrophobic residues in the TMD were changed to arginine.

RNCs with an intact TMD, either exposed or buried in the tunnel, efficiently recovered SRP as judged by immunoblotting for SRP54 (*Figure 1B*). Furthermore, the SRP68 and SRP72 subunits were recovered at sufficient levels to be clearly visible on the stained gel of the purified complexes (*Figure 1B*). Indeed, immunoblotting of the unbound fraction after affinity purification revealed that ~75% and 90% of SRP in the lysate was depleted by RNCs with the TMD in the tunnel or exposed, respectively (*Figure 1C*), despite these representing only ~5% of total ribosomes. Appreciable SRP depletion was not observed with either of the 3R RNCs, and post-translational TMD-binding proteins such as TRC40 (*Stefanovic and Hegde, 2007*) were not depleted by any of the constructs. These results illustrate that RNCs containing a TMD inside the ribosomal tunnel recruit SRP nearly as efficiently as RNCs containing an exposed TMD.

The observation that the 3R mutant largely abolishes SRP recruitment suggests that shorter hydrophobic regions, such as those remaining on either side of the mutation, are insufficient for robust recruitment. Serial truncations verified that maximal recruitment required the presence of 14 hydrophobic residues in the tunnel, with lower (but detectable) recruitment once 10 or 11 residues has been synthesized (*Figure 1D*). Analysis of shortened TMDs were consistent with this result, showing

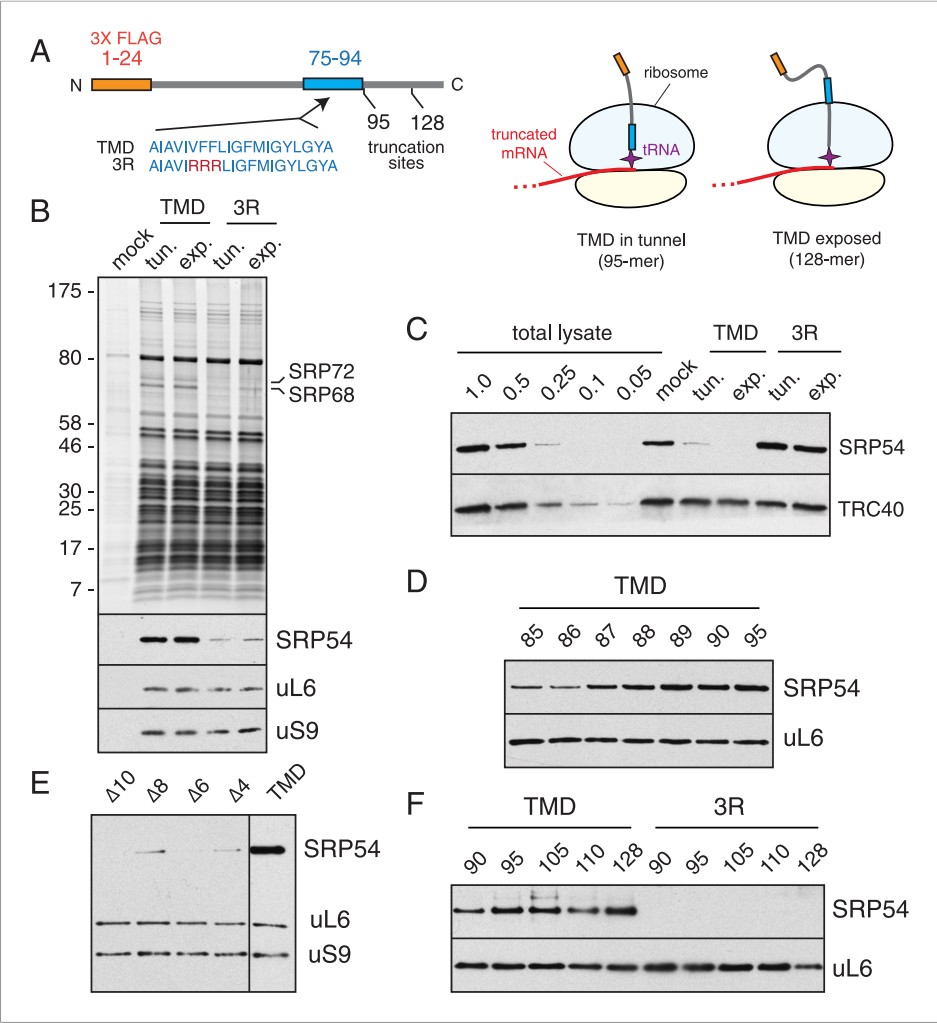

**Figure 1**. The timing of SRP recruitment to the mammalian ribosome. (**A**) Schematic of the constructs used for production and purification of stalled RNCs. Key domains and their positions (in amino acids) are indicated, along with the mutation (3R) used to disrupt the TMD. The diagrams to the right depict RNCs with the TMD inside or outside the tunnel used for biochemical and structural analyses. (**B**) Anti-FLAG affinity purifications were performed on translation reactions programmed with no RNA (mock), TMD-containing transcripts, or 3R mutant transcripts. Truncation was at position 95 (tunnel) or 128 (exposed). The samples were analyzed by SDS-PAGE and visualized for total protein (top panel) or immunoblotted for SRP54, the large ribosomal protein uL6, or the small subunit protein uS9. (**C**) The unbound fractions of translation reactions following affinity purification of the TMD and 3R constructs (as in panel **B**) were analyzed relative to serial dilutions of total lysate. RNCs containing an intact TMD, whether in the tunnel or exposed, selectively deplete more than 75% of SRP from the lysate. (**D**) RNCs truncated at different positions relative to the TMD were affinity purified and probed for SRP and the ribosome as in panel **B**. Maximal recruitment is observed at all points after truncation at residue 89, when 14 TMD residues have entered the tunnel. (**E**) Analysis of SRP recruitment as in panel **B** using mutant TMDs in which 4 to 10 residues have been deleted. The truncation point was at residue 95. (**F**) Experiment as in panel **D** for the indicated truncation points using the TMD or 3R construct.

sharply diminished recruitment with deletion of 4–8 residues, and no detectable recruitment after deletion of 10 residues (*Figure 1E*).

Systematically varying the position of the full TMD inside the tunnel showed efficient SRP recruitment at all positions (*Figure 1F*), while the shortened Δ8 mutant showed low recruitment at all positions (data not shown). These data collectively suggest that SRP binds translating ribosomes in two distinct states: a 'scanning' mode when the TMD is inside the exit tunnel, and an 'engaged' mode

where SRP interacts directly with the hydrophobic nascent chain. These scanning and engaged SRP-RNCs were assembled in total cytosol from endogenous mammalian components, thereby representing native complexes. Importantly, both complexes could be purified in sufficient amounts for structural analysis by electron cryomicroscopy (cryo-EM).

## Cryo-EM structures of SRP in its scanning and engaged states

Ribosomal particles from the scanning and engaged SRP complexes were visualized by cryo-EM and subjected to iterative in silico classification (*Figure 2—figure supplement 1*) to select RNCs containing both P-site tRNA (as a surrogate for the nascent chain) and SRP. Refinement of the resulting RNC populations resulted in final reconstructions at an overall resolution of 3.9 Å and 3.8 Å for the 80S ribosome bound to SRP in its scanning and engaged modes, respectively (*Figure 2A,B*, *Table 1*, *Figure 2—figure supplement 2*). The inherent flexibility of regions of SRP was reflected by the large variation in local resolution within the particle, which spans from ~3.5 Å to greater than 7.5 Å resolution (*Figure 2—figure supplement 3*).

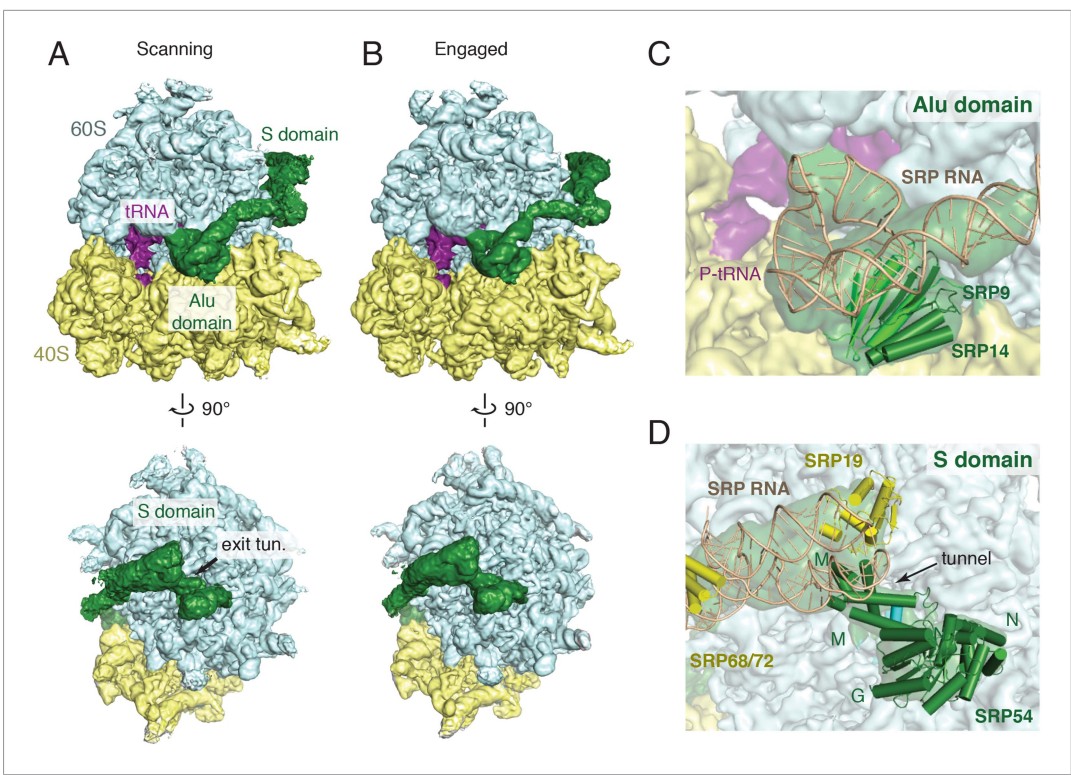

**Figure 2**. Overview of scanning and engaged SRP-RNC cryo-EM reconstructions. (**A**, **B**) Two views depicting the cryo-EM density of the scanning and engaged complex, in which the density for SRP is green, 40S is yellow, 60S is blue, and P-site tRNA is purple. The top and bottom panels show views of the GTPase center (with bound Alu domain) and the exit tunnel (with bound S domain), respectively. (**C**, **D**) Models for the Alu and S domains of SRP are overlayed on the appropriately colored density map for the engaged complex (yellow: 40S, blue: 60S, purple:tRNA, green: SRP). See also *Figure 2—figure supplements 1–4*.

The following figure supplements are available for figure 2:

**Figure supplement 1**. Schematic of computational classification.

**Figure supplement 2**. Map and model quality.

**Figure supplement 3**. Local resolution of cryo-EM maps.

**Figure supplement 4**. Nascent polypeptide inside the exit tunnel.

**Table 1**. Refinement and model statistics

| Data collection | Scanning | Engaged |
|---|---|---|
| Particles | 27,415 | 52,061 |
| Pixel size (Å) | 1.34 | 1.34 |
| Defocus range (μm) | 2.0–3.5 | 2.0–3.5 |
| Voltage (kV) | 300 | 300 |
| Electron dose (e/Å$^2$) | 27 | 27 |
| Map sharpening B-factor (Å$^2$) | −77.0 | −82.8 |
| | **60S-S domain** | **40S-Alu domain** |
| Model composition | | |
| Non-hydrogen atoms | 149,733 | 78,464 |
| Protein residues | 7571 | 5021 |
| RNA residues | 4257 | 1824 |
| Ions | 133 | 36 |
| Refinement | | |
| Resolution used (Å) | 3.8 | 3.8 |
| Average B factor ( Å$^2$) | 89.1 | 109.6 |
| R factor* | 0.30 | 0.32 |
| Fourier Shell Correlation (FSC)† | 0.85 | 0.79 |
| Rms deviations | | |
| Bonds (Å) | 0.011 | 0.011 |
| Angles (°) | 1.9 | 1.9 |
| Ramachandran plot | | |
| Favored (%) | 93 | 91 |
| Outliers (%) | 7 | 9 |

*R factor = $\Sigma||F_{obs}| - ||F_{calc}|/\Sigma|F_{obs}|$.
†FSC = $\Sigma(F_{obs}F^*_{calc})/\sqrt{(\Sigma|F_{obs}|^2 \ \Sigma|F_{calc}|^2)}$.

In both structures, the ribosome is unratcheted and contains a canonical P-site tRNA and density for the nascent chain in the exit tunnel (*Figure 2—figure supplement 4*). Density for the mRNA codon in the P site is also present, but is poorly defined, likely reflecting increased flexibility due to truncation of the mRNA. Additional density representing the E-site tRNA is also observed, though its anticodon is disordered suggesting it is not in a single orientation. We do not observe any conformational changes in the ribosome exit tunnel in the scanning structure relative to either the engaged structure or earlier mammalian ribosome structures (*Voorhees et al., 2014*). Thus, SRP recruitment by a TMD inside the exit tunnel cannot be explained purely based on structural changes observed at this resolution, a point we address further in the 'Discussion'.

The overall architecture of SRP in the engaged binding mode (*Figure 2B*) is very similar to earlier reconstructions of this state visualized using heterologous plant–mammalian complexes (*Halic et al., 2004*, *2006*). As expected, the Alu domain is localized to the GTPase center between the 40S and 60S subunits (*Figure 2C*) and is connected by a flexible RNA linker to the S domain positioned at the exit tunnel (*Figure 2D*). At low-resolution, the observed density accounts for the majority of the SRP molecule, including most of the RNA, the Alu domain proteins (SRP9 and SRP14), and most of the S domain proteins (SRP54, SRP19, and RNA binding regions of SRP68). However, only the Alu domain and SRP54 were sufficiently well ordered to allow direct modelling into the density maps, and thus form the focus of our interpretations in the sections below. The others areas, whose functional relevance remain poorly understood, were rigid-body fit with minor adjustments from available crystal structures (*Grotwinkel et al., 2014*), and were not interpreted further here.

The SRP density in the matched scanning mode structure was found to be remarkably similar in overall conformation, position, and relative occupancy of all domains (*Figure 2A*). In particular, the Alu domain, whose binding has been implicated in slowing translational elongation after SRP engages with a signal sequence (*Wolin and Walter, 1989*), is positioned in the GTPase center. Similarly, the entire S domain is grossly indistinguishable, indicating that the TMD-interacting M domain of SRP54 is positioned at the exit tunnel even prior to engagement. This suggests that in mammals, SRP has a single stable binding site on the ribosome, and carries out its biological functions during both scanning and engagement from this position.

## Analysis of SRP interactions with the ribosome

The considerably higher resolution of these structures relative to earlier reconstructions permitted interpretation of several SRP-RNC interactions in atomic terms. SRP is anchored on the ribosome primarily by its SRP54 subunit near the exit tunnel (*Figure 3A*) and the Alu domain at the ribosomal GTPase center. Two loops in the SRP54 N-domain previously implicated in ribosome binding (*Halic et al., 2004*, *2006*; *Schaffitzel et al., 2006*) can now be sufficiently resolved for unambiguous placement of many protein side chains and analysis of the stabilizing chemical interactions (*Figure 3B*). For example, Thr21 and Asp19 of SRP54 are within hydrogen bonding distance of Lys46 in uL29 and the phosphate oxygen of A80 in 5.8S rRNA (*Figure 3C*). Similarly, the backbone of residue 67 is within hydrogen bonding distance of the carbonyl oxygen of residue 116 of uL23 (*Figure 3D*). These specific contacts, along with other electrostatic interactions involving the conserved basic residues in this region, provide the primary stabilization of SRP54 and serve to orient its signal-binding M domain at the ribosomal exit tunnel.

Within the ribosomal GTPase center, the RNA portion of the well resolved Alu domain approaches the 60S subunit, interacting with the ribosomal stalk protein uL10, while the proteins SRP14 and SRP9

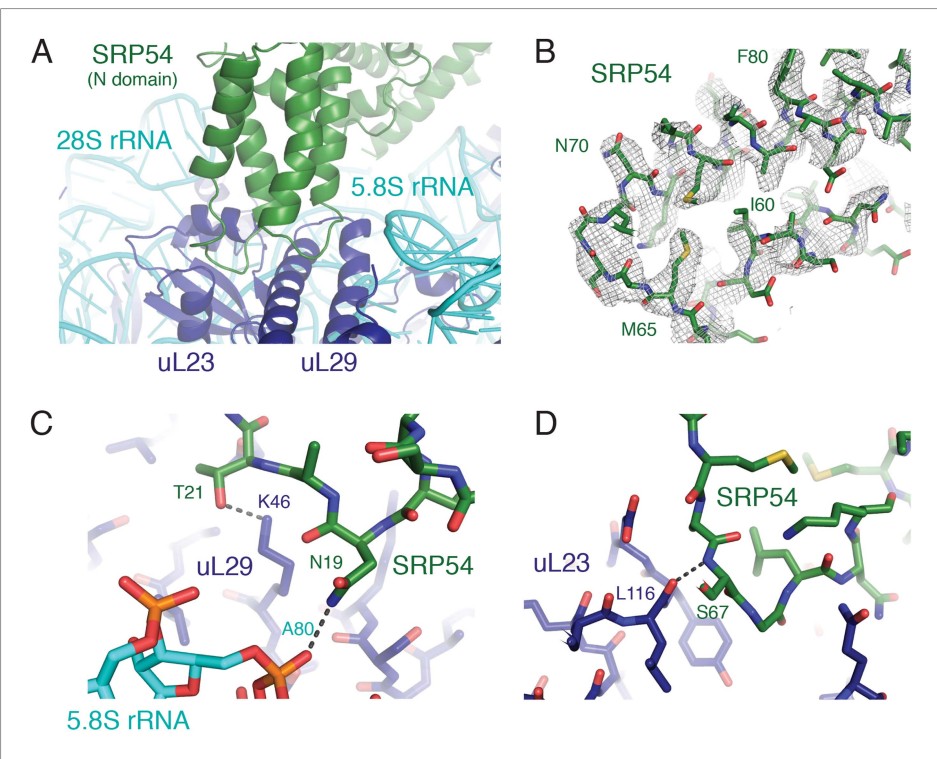

**Figure 3**. SRP-ribosome interactions near the exit tunnel. (**A**) Overview of the N-domain of SRP54 (green), which is positioned near the exit tunnel by its interactions with the 5.8S rRNA and ribosomal proteins uL23 and uL29. (**B**) Representative density for one of the loops of SRP54, which anchors SRP to the ribosome, and its adjoining helices. (**C**, **D**) Specific hydrogen bonding interactions of SRP54 with the 5.8S rRNA (cyan) and ribosomal proteins uL29 and uL23 (dark blue).

approach the 40S subunit, interacting with the backbone of the 18S rRNA (*Figure 4A*). The structure reveals that Asn77 and Lys95 of SRP14 are positioned within hydrogen bonding distance of the phosphate oxygens of U488 and A464 of the 18S rRNA (*Figure 4B–D*). Of note, alignment of our SRP structure bound to the mammalian ribosome with an earlier structure of mammalian SRP bound to the plant ribosome (*Halic et al., 2004*) suggests changes in relative position of the S and Alu domains. Such changes, together with sequence divergence of uL10 and uL11, may explain why mammalian SRP arrests translation in a plant system while only subtly slowing translation in mammals (*Walter and Blobel, 1981*; *Wolin and Walter, 1989*).

## The SRP54 M-domain before and after substrate engagement

The main functional transition between a scanning and engaged SRP is substrate binding by the M domain. Our structures afforded the first opportunity to visualize the conformation of the scanning M domain and deduce changes that accompany TMD engagement. We began by modelling the engaged M domain (*Figure 5A*), guided by earlier crystal structures involving this region (*Keenan et al., 1998*; *Clemons et al., 1999*; *Batey et al., 2000*; *Rosendal et al., 2003*; *Egea et al., 2008*). The hydrophobic groove of this domain is occupied by density consistent with an alpha helix of ~12 residues, accounting for roughly half of the nascent chain's TMD. This assignment is consistent with earlier structures (*Janda et al., 2010*; *Hainzl et al., 2011*) and crosslinking analyses (*Zopf et al., 1990*; *Lutcke et al., 1992*).

After fitting the density corresponding to earlier crystal structures and assigning the substrate, two additional regions of helical density were observed abutting the TMD (*Figure 5A*). Several arguments suggest that these represent the C-terminus of SRP54. First, the C-terminal ~70 residues, truncated from all earlier SRP54 crystal structures, are the only unaccounted amino acids in this region. The only other nearby polypeptide would be the hydrophilic nascent chain, which is unlikely to be helical or rigid. Second, this 70 residue stretch of mammalian SRP54 contains two conserved predicted helices of

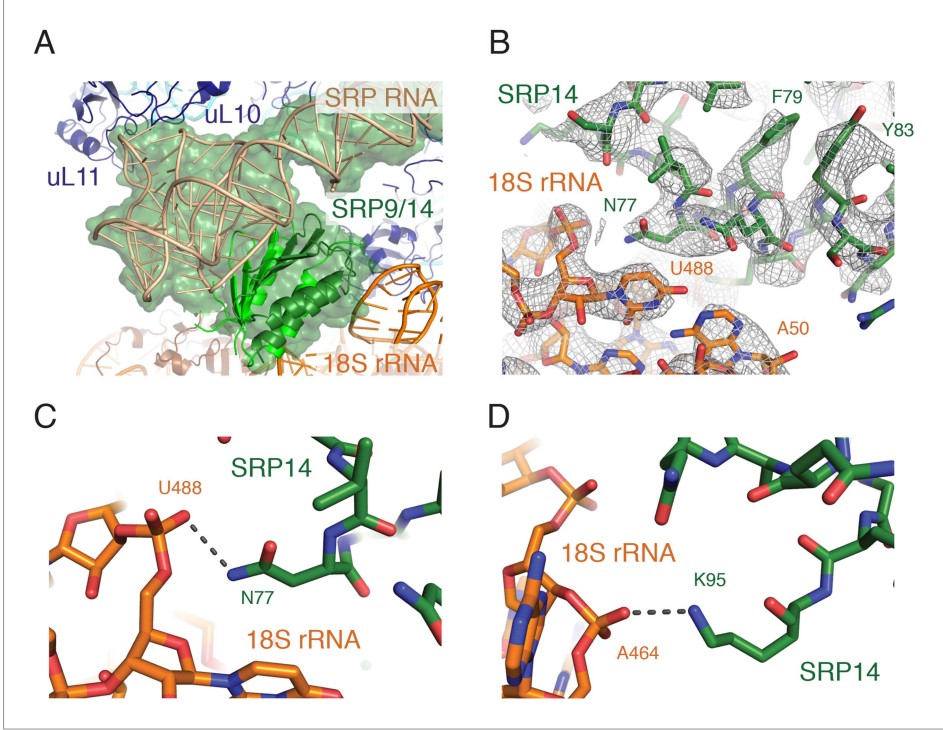

**Figure 4**. Interactions of the SRP Alu domain at the GTPase center. (**A**) Overview of the Alu domain (green) contacting the 60S subunit via the ribosomal stalk (dark blue) and the 40S subunit through the 18S rRNA (orange). The engaged structure is shown; the scanning structure was essentially indistinguishable. (**B**) Representative density for the region of SRP14 that interacts with the 40S subunit. (**C**, **D**) Specific hydrogen bonding interactions of SRP14 with the backbone of the 18S rRNA.

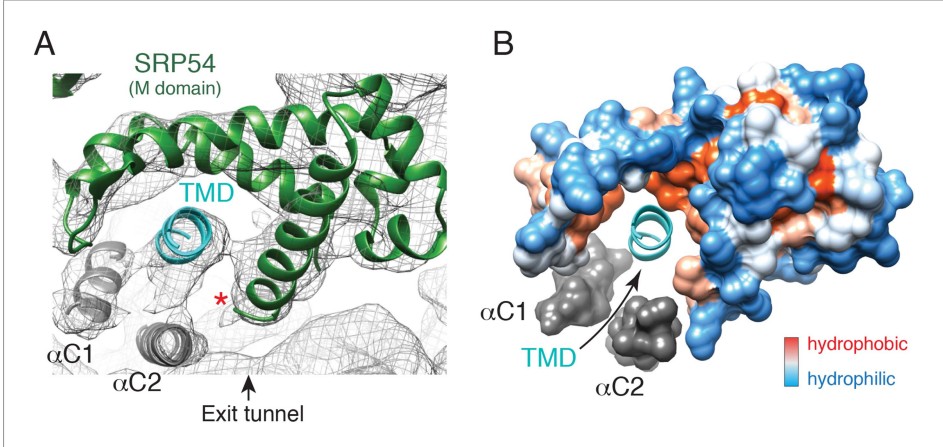

**Figure 5**. The engaged SRP54 M domain. (**A**) The final model of the SRP54 M-domain (green) and bound TMD (cyan) is displayed in the density for the engaged complex. The C-terminus of this domain is indicated with a red asterisk. Density for two additional helices (termed αC1 and αC2), which were not part of the earlier crystal structures, represent the C-terminus of SRP54 and would be connected via flexible linkers (not modelled) to the remainder of the M-domain. These helices enclose the TMD substrate, minimizing the exposed hydrophobic surface area. (**B**) Displayed is a surface representation of the M-domain, colored by hydrophobicity, indicating the hydrophobic binding pocket for the TMD enclosed by the C-terminal helices of SRP54. See also *Figure 5—figure supplement 1*.

The following figure supplement is available for figure 5:

**Figure supplement 1**. Sequence conservation of the amphipathic helices at the C-terminus of SRP54.

appropriate length (*Figure. 5—figure supplement 1A*). Third, these helices are amphipathic, making them ideal for surrounding the hydrophobic TMD (*Figure 5—figure supplement 1B*). We therefore posit that these two additional helices, provisionally termed αC1 and αC2 of SRP54, serve as an intramolecular 'lid' that surrounds the hydrophobic substrate to shield it from aqueous solvent (*Figure 5B*). Although we cannot unambiguously assign these helical regions to a particular sequence, we have provisionally named them based on the relationship between their apparent length in the structure and predicted size.

Relative to the engaged state, the scanning structure showed markedly underrepresented density in the region of αC2, but clear helical density within the SRP hydrophobic groove (*Figure 6A*, compare top and bottom panels). This suggests that in the absence of a TMD substrate, αC2 shifts to occupy the hydrophobic groove. This helix may be somewhat dynamic since a small amount of density is still observed in the alternate position of αC2 (*Figure 6A*, bottom panel). Although this is our favored hypothesis, we cannot formally exclude the possibility that the density in the hydrophobic groove represents the nascent chain. However, this seems unlikely because a hydrophilic sequence would be disfavored in the hydrophobic groove, this region of polypeptide is unlikely to be helical, earlier crosslinking data suggest SRP does not interact with hydrophilic segments of nascent chain (*Berndt et al., 2009*), and disappearance of the αC2 density would remain unexplained. Thus, we propose that the amphipathic αC2 normally resides in the hydrophobic groove, but is displaced by TMD engagement to its 'lid' position between the M domain and exit tunnel (*Figure 6B*).

## Analysis of SRP-ribosome interactions during translation elongation

Our structures indicate that Alu domain occupancy at the GTPase center is mutually exclusive with translational GTPase interactions that occur with each cycle of nascent chain elongation. Yet, 35 cycles of translational elongation separate our two structures, and SRP does not arrest translation even after successful engagement (*Wolin and Walter, 1989*). To investigate this apparent incompatibility, we sought to examine the relationship between translational GTPase binding and the SRP-ribosome interaction in the scanning vs engaged modes. Such an experiment is complicated because neither dominant negative eEF1 nor eEF2 would bind our staged SRP-RNC complexes: eEF1 ternary complex binding requires an appropriate codon in the ribosomal A site, while eEF2, when in its GTP-bound state,

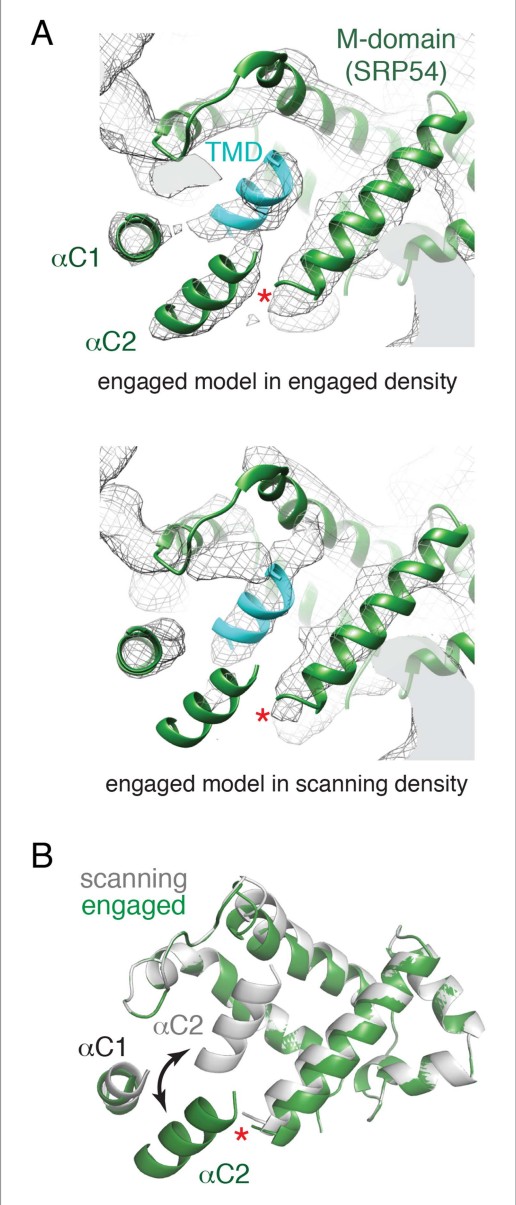

**Figure 6**. Comparison of the engaged and scanning M-domain. (**A**) Density and associated model of the engaged M-domain (top panel) and rigid-body fitting of the engaged model into the density for the scanning structure (bottom panel). Note that density in the engaged position of αC2 is nearly absent in the scanning structure. The C-terminus of the M-domain preceding αC1 and αC2 is indicated by a red asterisk. (**B**) Superposition of the scanning and engaged M-domain models suggests that αC2 is repositioned (arrow) upon binding to a TMD (not depicted for clarity).

preferentially acts on pre-translocation RNCs in the hybrid state. Furthermore, adding mutant versions of these factors to our reactions would inhibit translation. To bypass these technical obstacles, we utilized the GTPase Hbs1 as a surrogate for eEF1 and eEF2. The G-domain of all three factors share a high degree of sequence and structural homology, and the binding of Hbs1 to the ribosomal GTPase center is well characterized (*Shoemaker et al., 2010*; *Becker et al., 2011*; *Pisareva et al., 2011*). Importantly, a GTPase deficient dominant negative mutant (Hbs1-DN) added to our translation reactions will stably bind to the translation factor binding site (*Becker et al., 2011*) only when ribosomes reach the end of the truncated message (*Shao and Hegde, 2014*), allowing us to query the consequence of GTPase center occupancy for the SRP-RNC interaction without inhibiting translation.

Translation extracts were supplemented with increasing amounts of Hbs1-DN, and used for the synthesis and purification of scanning and engaged RNC complexes via the nascent chain. Hbs1-DN binding, observed in both cases, reduced SRP recovery by up to ~70% with the scanning RNCs, but had no effect on engaged SRP-RNC complexes (*Figure 7A*). Because a substantial excess of Hbs1-DN relative to ribosomes was used, its failure to displace engaged SRP suggested the existence of a ternary complex between the ribosome, Hbs1-DN, and SRP. The relatively modest displacement of scanning SRP further hinted that this too may permit ternary complex formation.

To test this idea, scanning and engaged HA-tagged RNCs were assembled in the presence of FLAG-tagged Hbs1-DN, affinity purified via the FLAG tag, and tested for SRP recovery (*Figure 7B*). Consistent with the competition experiment, equal amounts of ribosomes were recovered from both reactions in an Hbs1-dependent manner, but substantially less SRP was found with scanning vs engaged RNCs. Thus, when SRP is in its scanning mode, binding of Hbs1-DN can partially (but not completely) displace SRP. However, once the TMD is exposed, both Hbs1-DN and engaged SRP are able to stably associate with the ribosome.

Such an observation can be explained by the available structural information. Due to the flexible RNA linker that connects the two halves of SRP, Alu domain dissociation need not lead to immediate dissociation of SRP. Rather, the SRP S domain interactions could retain SRP on the ribosome, with dissociation governed by the off rate of this complex. For *E. coli* SRP, which lacks an Alu domain, this off rate is ~0.1–0.3 s$^{-1}$ (*Saraogi et al., 2014*), probably explaining why scanning-mode SRP is lost during the 2–3 hr needed for RNC affinity purification. By contrast,

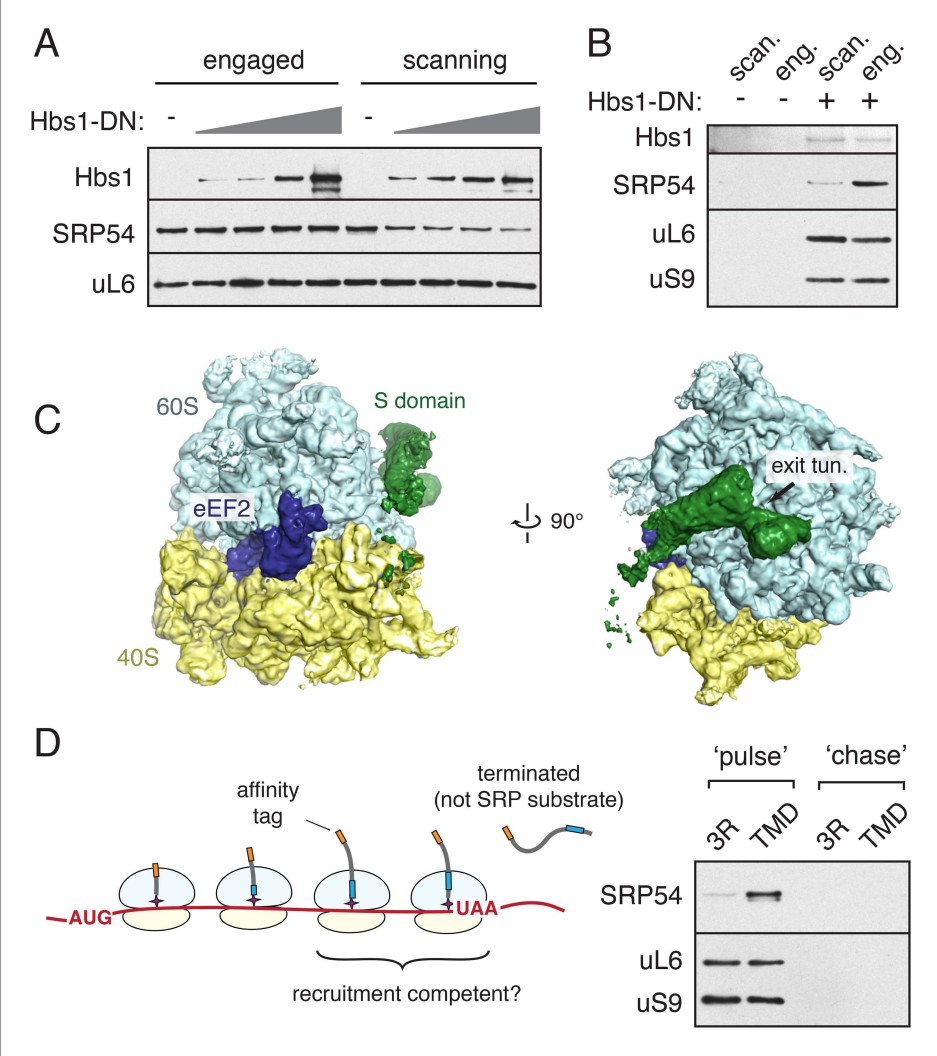

**Figure 7**. Integration of SRP recruitment with the translational elongation cycle. (**A**) Stalled RNCs were produced and affinity purified via the nascent chain from translation reactions supplemented with increasing amounts of a GTPase deficient Hbs1 (Hbs1-DN). SRP recovery is reduced by Hbs1-DN for scanning mode RNCs, but unaffected for engaged RNCs. (**B**) Following translation in the presence of 3X-FLAG tagged Hbs1-DN, HA-tagged RNCs were purified via the FLAG tag and analyzed for the indicated components. Recovery of SRP in a ternary complex with Hbs1 and the ribosome is observed for both the scanning and engaged mode RNCs, but SRP recovery is substantially reduced in the scanning mode. (**C**) Two views depicting the cryo-EM density for a ternary complex of the ribosome bound to eEF2 (blue) in the GTPase center and the S domain of SRP (green) at the exit tunnel. (**D**) Left panel: diagram illustrating that translation of a construct containing a stop codon 35 amino acids from the beginning of the TMD will only expose the TMD after termination. Right panel: the construct from the diagram (or a 3R control) was translated for a short 3 min 'pulse' followed by either immediate cooling, or addition of an initiation inhibitor and a 30 min 'chase' translation. Affinity purification of the nascent chain from the pulse and chase samples reveals that actively translating RNCs can recruit SRP in a TMD-dependent manner. See also *Figure 7—figure supplement 1*.

The following figure supplement is available for figure 7:

**Figure supplement 1**. Simultaneous binding of eEF2 and SRP.

the additional interaction between the TMD and the M domain reduces this off rate effectively to zero relative to the timeframe of this experiment, explaining why Hbs1-DN cannot displace engaged SRP. Other translational GTPases (eEF1, eEF2, and the release factor complexes)

would likely share the same qualitative relationship with the different states of ribosome-bound SRP, explaining why SRP, even after RNC engagement, is compatible with active elongation.

This logic suggests that a population of ribosomes bound to both a translation factor and SRP might be present in our EM samples. The use of a truncated mRNA for sample preparation precludes eEF1 binding, but eEF2-ribosome-SRP complexes were feasible. Such a population was not identified in our scanning dataset, likely due to the lower stability of this ternary complex when SRP is bound in its scanning mode (*Figure 7B*). By contrast, the engaged dataset contained a small subset of particles in which density was observed for both eEF2 and the S domain of SRP bound in its canonical location at the exit tunnel (*Figure 7C*). At a lower threshold, density for the flexible RNA linker and the Alu domain could also be observed, with the Alu domain at the so-called right foot of the 40S subunit (*Figure 7*, *Figure 7—figure supplement 1*). The relatively poor density for this half of SRP suggests it is probably dynamic when the Alu domain cannot bind its canonical site at the GTPase center. Whether the weakly observed alternate Alu domain position on the 40S represents a physiologically relevant binding site remains to be determined. Nevertheless, this structure supports our conclusion that the Alu domain can be displaced from the GTPase center by translation factors without dissociating SRP from the ribosome.

The compatibility of SRP and translation factor binding to the same ribosome suggested that scanning SRP need not dissociate between rounds of polypeptide elongation. To investigate this idea, we designed an experiment to determine if scanning SRP can be observed on actively translating ribosomes. A stop codon was positioned 35 residues from the beginning of the TMD, ensuring that translation would terminate before emergence of the TMD from the exit tunnel (*Figure 7D*, diagram). During translation, a sub-population of ribosomes containing nascent peptides long enough to expose the N-terminal epitope tag would also contain the TMD inside the tunnel. As SRP cannot interact with its clients post-translationally, its recovery with the nascent chain would necessarily be via the ribosomal scanning mode.

Affinity purification of RNCs after 3 min of translation recovered substantially more SRP with RNCs containing a TMD than the 3R mutant (*Figure 7D*, right panel). Parallel reactions in which translation initiation was inhibited at 3 min and the ribosomes were allowed to complete translation produced full-length protein that recovered neither ribosomes nor SRP. These observations suggest that SRP is recruited in a TMD- and nascent chain-dependent manner to actively translating ribosomes despite never exposing the TMD outside the exit tunnel. Thus, SRP might be able to continuously scan translating ribosomes as a TMD elongates through the exit tunnel, thereby permitting engagement with essentially no opportunity for TMD exposure to the cytosol.

## Discussion

In this study, we have characterized an anticipatory scanning mode of SRP-ribosome interaction, validated its existence during ongoing translation, and provided the first structure of this complex. Together with our engaged SRP complex, these structures represent markedly improved views of many functional regions of mammalian SRP and provide the first entirely homologous SRP-RNC structures assembled with native endogenous factors. Additional experiments analysing the relationship between SRP and a translational GTPase permit us to consolidate our findings into a dynamic model of nascent chain scanning and TMD capture by SRP during the earliest stages of membrane protein biosynthesis (*Figure 8*).

Once as few as 12–14 residues of a TMD enter the ribosomal exit tunnel, an RNC becomes competent to recruit SRP in its scanning mode. SRP binding appears to favor an unratcheted ribosome and requires an unoccupied GTPase center. An A site tRNA, while not observed in our structure due to the use of a truncated mRNA, is sterically compatible with SRP binding. Thus, Alu domain recruitment could potentially occur immediately after dissociation of eEF1 or eEF2, from the ribosome. The S domain could presumably bind transiently anytime, and may increase local SRP concentration to facilitate the Alu domain interaction. The proportion of each ~200 ms translation cycle during which RNCs are compatible for Alu domain binding remains to be determined. Regardless, a twenty residue TMD would have approximately 21 translation cycles to recruit SRP while it is still inside the tunnel.

Once recruited, SRP would be ribosome-associated via both the S and Alu domains. Arrival of the next translation factor would displace the Alu domain, while the S domain can remain fixed due to the flexible RNA linker between these two domains (*Figure 8A*). Whether SRP completely dissociates

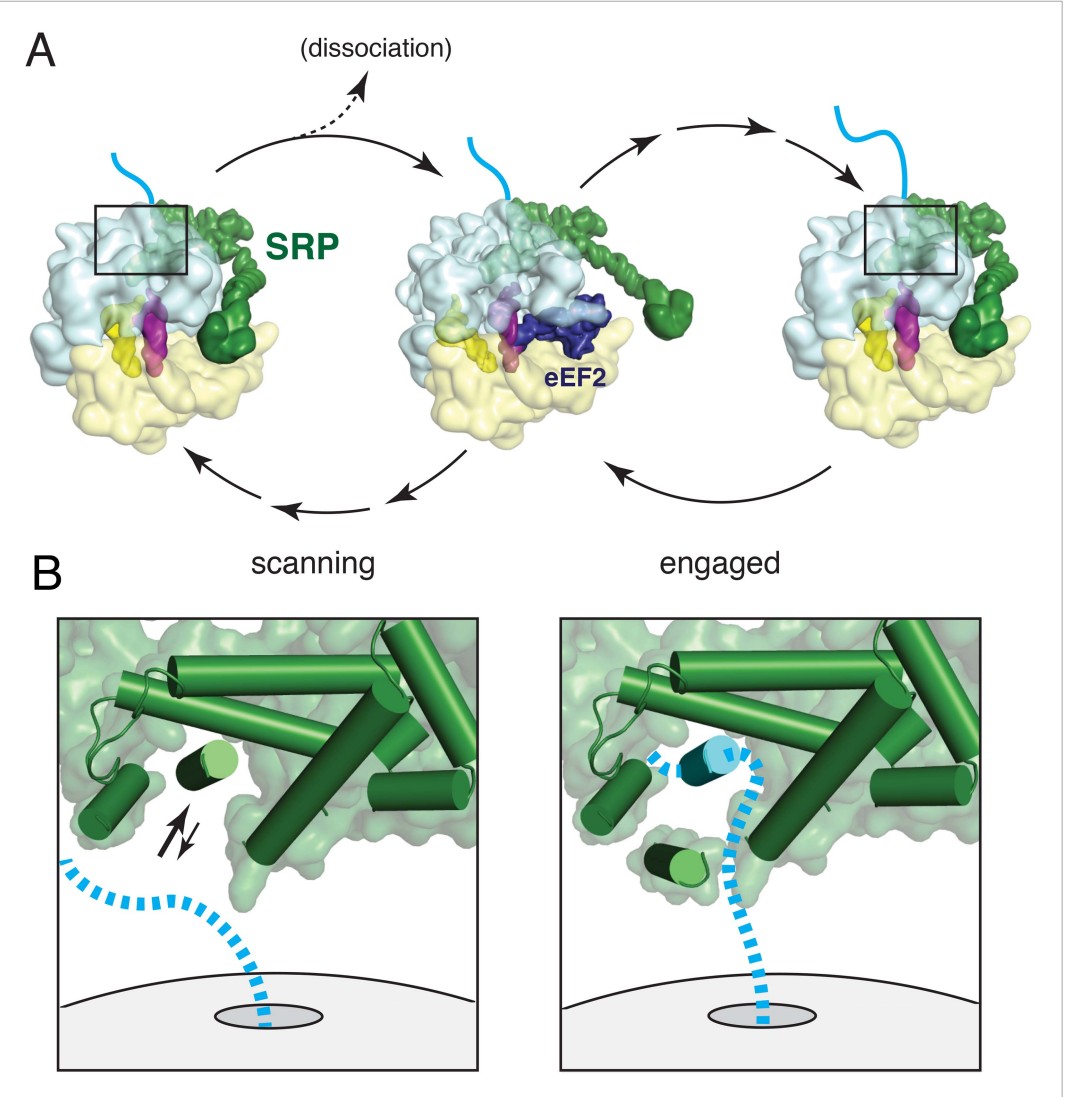

**Figure 8**. Working model for SRP scanning and engagement of a nascent membrane protein. (**A**) Model for SRP dynamics and conformation during translational elongation in its scanning and engaged states. The Alu domain swings away from the ribosome to accommodate translation factor binding (eEF2 is shown). Scanning SRP is more prone to displacement by translation factors, while engaged SRP remains stably bound. (**B**) Close-up views of the relative positions and conformations of the SRP54 M-domain (green) and hypothetical nascent chain (blue) during scanning (left) and engagement (right). The arrows in the scanning diagram depict hypothesized dynamics of the αC2 helix to permit nascent chain sampling. A hydrophobic domain (blue cylinder) displaces the αC2 helix to its lid position. See also *Figure 8—figure supplement 1*.

The following figure supplement is available for figure 8:

**Figure supplement 1**. Superposition of various M-domain structures.

from the ribosome would depend on the off-rate of the S domain relative to the next Alu domain binding opportunity within the translation cycle. Estimates from the *E. coli* system suggest that the S domain dissociation would be far slower than an elongation cycle, and thus the Alu domain could re-bind and detach repeatedly while SRP nevertheless maintains ribosome association during ongoing translation. This would explain how we are able to isolate scanning mode SRP-RNC complexes from an active translation reaction (*Figure 7D*) and recover at least some GTPase-SRP-RNC ternary complex in the scanning state (*Figure 7B*).

Early recruitment followed by maintenance of the M domain at the exit tunnel would allow for continuous sampling of the nascent chain until the TMD emerges. The M domain is optimally pre-positioned for this function, with the hydrophobic groove located ~20 Å from the exit site. Although the precise path of the nascent chain is not known, the limited space in this region means it necessarily passes the M domain (*Figure 8B*, left panel). We envisage a dynamic αC2 helix that transiently exposes the hydrophobic groove to provide opportunities for nascent chain sampling. If a segment of the nascent polypeptide is sufficiently hydrophobic to replace the αC2 helix, it would successfully engage SRP54 (*Figure 8B*, right panel). In this model, the biophysical properties of αC2 effectively set the hydrophobicity threshold for client engagement. After nascent chain binding to the M domain, αC2 would occupy a position on the opposite side of the hydrophobic groove, oriented with its hydrophobic face toward the substrate, forming part of a protective lid. The conformation of this engaged state may be flexible to accommodate different substrates, as suggested by other variant configurations (*Figure 8—figure supplement 1*). Regulated access to a hydrophobic binding groove and substrate shielding by an amphipathic helix appears to be a general principle that is shared by the Get3 targeting factor (*Mateja et al., 2015*).

In the engaged state, the SRP S domain is able to remain stably bound to the ribosome via its dual interactions with the exit tunnel and nascent chain, ensuring shielding of the hydrophobic substrate. Elongation factors could still access the GTPase center by displacing the Alu domain, which must swing away from the ribosome. Its local concentration would necessarily remain high due to S domain tethering, presumably improving its ability to compete with eEF1 and eEF2. This is the likely cause of the reported slowing of translation by SRP (*Wolin and Walter, 1989*) that provides a subtly increased time for successful delivery to the translocon (*Mason et al., 2000*; *Lakkaraju et al., 2008*).

It is noteworthy that the minimum hydrophobic element for robust early recruitment of SRP is longer than the 7 to 9 hydrophobic residues within most cleavable N-terminal signal peptides (*von Heijne, 1990*). Early recruitment may not be critical in this case because signal peptides are more soluble and less aggregation-prone than TMDs, and sufficiently short to be fully shielded by the hydrophobic groove of the M domain. By contrast, TMDs responsible for targeting are markedly more hydrophobic and between 17 to 24 residues long. They are effectively insoluble in aqueous solvent and too long to be completely shielded by the M domain, making their efficient and early targeting more important. Moreover, early recruitment would provide SRP the opportunity for first refusal of an emerging TMD, which could allow SRP to out-compete other more abundant binding factors such as TRC40 (*Stefanovic and Hegde, 2007*), SGTA (*Wang et al., 2010*), Hsp70 (*Rabu et al., 2008*), and calmodulin (*Shao and Hegde, 2011a*).

Another important functional implication of early recruitment is increased time for targeting the RNC to the translocon. We cannot, at this level of resolution, see any differences in the configuration of the SRP54 NG domain between the scanning and engaged states. This suggests that the scanning state should be competent for interaction with the SRP receptor, potentially permitting targeting while the TMD is still inside the tunnel. Based on the ~35 residue length of the exit tunnel, recruitment before the TMD is synthesized provides an additional ~7 s for targeting relative to a model where targeting only occurs after TMD emergence.

This extra time would be sufficient for delivery to the translocon before or as the TMD emerges, essentially eliminating its cytosolic exposure and allowing its synthesis to occur directly at Sec61. Indeed, recent ribosome profiling experiments in yeast suggest that many proteins arrive at the Sec61 translocon before the TMD is exposed (*Jan et al., 2014*). Such early delivery may be particularly important for proteins whose downstream domains may interfere with correct translocation such as downstream TMDs of multispanning membrane proteins or rapidly folding domains (*Conti et al., 2014*). Our findings suggest that early SRP recruitment is another mechanism for lengthening the targeting window, and may operate in conjunction with translational slowing (*Wolin and Walter, 1989*) and strategic positioning of slowly decoded rare codons (*Pechmann et al., 2014*).

A major unresolved question is the mechanistic basis for how a TMD can influence SRP binding from inside the exit tunnel. One possibility involves conformational changes on the ribosome surface triggered by TMD interactions with the proteins or RNA lining the tunnel. However, we have been unable to detect any differences in the tunnel or surface regions for SRP binding by comparing our scanning RNCs with those containing hydrophilic sequences. Instead, we currently favor a kinetic model in which a hydrophobic sequence in the tunnel biases ribosomes into a conformation favorable for Alu domain binding. If the presence of a hydrophobic sequence in the ribosomal tunnel would

preferentially extend those stages of the translation cycle compatible with SRP binding, this could provide a mechanism for tunnel-initiated SRP recruitment.

Evidence for preferential SRP binding to some but not other states of the ribosome comes from experiments using different translation elongation inhibitors (*Ogg and Walter, 1995*). Cycloheximide, an elongation inhibitor that causes ribosomes to pause in a presumably unratcheted state, can rescue translocation defects caused by limiting SRP availability. By contrast, another elongation inhibitor anisomycin, which pauses ribosomes at the ratcheted state, does not rescue under similar conditions. This implies that situations preferentially favoring the cycloheximide state may permit SRP recruitment. Thus, it is plausible that a TMD inside the tunnel, or elongation of hydrophobic amino acids, would subtly influence the translation cycle to favor this state. Indeed, sequences inside the ribosomal tunnel are known to influence the translation cycle (*Seidelt et al., 2009*; *Bhushan et al., 2011*; *Wilson and Beckmann, 2011*), while the biophysical characteristic of the aminoacyl-tRNA in the A site influences the time spent in different ribosome states (*Lareau et al., 2014*). A model invoking the Alu domain would also explain why sequence-independent scanning in the *E. coli* system appears to operate differently than the TMD-dependence observed in eukaryotes.

Finally, it is worth noting that tools are now readily available to analyze early recruitment of SRP in vivo. The combination of ribosome profiling (*Ingolia et al., 2009*) together with selective retrieval of SRP complexes (*del Alamo et al., 2011*) should permit the mapping of all SRP-containing RNCs. Such analyses can provide a global view of not only SRP substrates in vivo, but also the precise timing of its recruitment for different types of cargos. We anticipate that, based on our analyses in the mammalian system, early recruitment should be seen for proteins with the most hydrophobic targeting sequences. Such proteins are also the most aggregation-prone when targeting fails, perhaps providing a strong selection pressure for evolution of an early targeting mechanism.

## Materials and methods

### Plasmids and antibodies and recombinant protein purification

An SP64 vector-based construct encoding the transferrin receptor TMD (AIAVIVFFLIGFMIGYLGYA) inserted into Sec61β (*Hessa et al., 2011*) was modified to contain an N-terminal affinity tag (3xFLAG or 3xHA). Phusion mutagenesis was used to generate the following TMD mutants: 3R (AIAVIRRRLIGFMIGYLGYA), Δ4 (AVIVFFLIGFGYLGYA), Δ6 (AIVFFLIGGYLGYA), Δ8 (AVFFLIGYLGYA), and Δ10 (AFFLGYLGYA). A stop codon was introduced by Phusion mutagenesis at position 110 for the experiment in *Figure 7D*. The mammalian expression construct for Hbs1-DN has been described (*Shao et al., 2013*), and was purified as before (*Shao and Hegde, 2014*). Antibodies against uL6 (anti-L9) and uS9 (anti-S16) were from Santa Cruz Biotechnology (Dallas, TX). Anti-SRP54 was from BD Biosciences (San Jose, CA). The antibody against TRC40 has been described (*Stefanovic and Hegde, 2007*). Anti-Flag and HA resin and 3X Flag and HA peptides were obtained from Sigma (St. Louis, MO).

### In vitro transcription, translation, and affinity purifications

Preparation and purification of stalled RNCs was performed as previously described (*Shao et al., 2013*). Briefly, the template for in vitro transcription was prepared by PCR from the constructs above using a 5′ primer just preceding the SP6 promoter, and a 3′ primer at the desired site of truncation (or downstream of the stop codon for full length products). All 3′ primers for truncations encoded a final valine codon, whose peptidyl-tRNA is least labile to hydrolysis (*Shao et al., 2013*). PCR products were purified and used for in vitro transcription and translation in rabbit reticulocyte lysate as described (*Sharma et al., 2010*). Translations were for were for 20–25 min at 32°C. For *Figure 7A*, Hbs1-DN was included at 3.3 nM, 17 nM, 33 nM and 133 nM. For *Figure 7B*, Hbs1-DN was included at 133 nM. For *Figure 7D*, actively translating, unstalled complexes were produced by allowing 3 min of translation at 32°C followed by either a rapid fivefold dilution in chilled buffer 1 (50 mM HEPES pH 7.5, 200 mM KAc, 15 mM MgAc$_2$, and 1 mM DTT), or addition of 70 µM aurin tricarboxylic acid (ATA) for the remainder of a 30 min translation before dilution.

All samples were then affinity purified via either the nascent chain or the tag on Hbs1-DN using anti-Flag or HA resin. The Flag tag was used on the nascent chain in all experiments except those that included Hbs1-DN, which instead used HA-tagged nascent chains. For biochemical analysis, the affinity resin was either added directly to undiluted translation reactions, or after dilution in buffer 1. The ratio of affinity resin to translation reaction was typically 1:50. Binding was performed in batch at

4°C for ~1–2 hr with gentle mixing, transferred to a micro-spin column, washed with ~25 vol (relative to resin) of buffer 1, and eluted for 30 min at 22°C in the same buffer supplemented with 0.2 mg/ml of the appropriate peptide. Samples for structural analysis were washed as above in buffer 1 containing an additional 200 mM or 400 mM KAc for the scanning and engaged complexes, respectively. Samples were eluted in buffer 1 supplemented with 3 mM GDPCP, concentrated by centrifugation (50,000 rpm for 75 min in a TLA55 rotor), and resuspended in a volume of buffer 1 to achieve ~125 nM ribosomes in the presence of 100 µM GDPCP.

## Grid preparation and data collection

Purified samples were applied to glow-discharged holey carbon grids (Quantifoil R2/2), which had been coated with a ~70 Å thick layer of amorphous carbon. Using an FEI Vitrobot, 3 µl of sample was applied to the grid, followed by a 30 s incubation at 4°C, 3 s of blotting, and flash-cooling in liquid ethane. Data were collected on an FEI Titan Krios at 300 KV using FEI's automated single particle acquisition software and defocus values of 2–3.5 µm. Images were recorded using a back-thinned FEI Falcon II detector at a calibrated magnification of 104,478 (pixel size of 1.34 Å). Individual frames from the detector were recorded as previously described (*Bai et al., 2013*).

## Image processing

Contrast transfer function parameters were estimated using CTFFIND3 (*Mindell and Grigorieff, 2003*), and micrographs that had evidence of astigmatism or drift were discarded. All automated particle picking, 2D and 3D classifications, and refinements were performed using RELION as described below (*Scheres, 2012*). Unsupervised 2D class averaging was used to discard any non-ribosome particles, resulting in a combined 326,981 and 639,184 particles for the engaged and scanning samples, respectively. Iterative rounds of 3D classification were then utilized to identify the population of ribosomes bound to SRP (*Figure 2—figure supplement 1*).

For the scanning sample, 27% of selected 80S particles (171,143) were computationally classified as being in an unratcheted conformation and bound to a P-site tRNA. As weak density for SRP in this population could be observed, a further focused classification was performed utilizing a mask around the observed extra-ribosomal density to identify 16% of this unratcheted population (27,627 particles) that contained SRP. An additional focussed classification using a mask around the P-Site tRNA identified a final 27,415 particles that are unratcheted, bound to SRP, and contain improved density for the nascent chain. The somewhat weaker density observed for the nascent chain in the scanning complex (*Figure 2—figure supplement 4*) might be due to either increased flexibility of the nascent chain when it is not anchored at the N-terminus by an interaction with SRP, or decreased occupancy resulting from hydrolysis during sample freezing that was not resolved by computational sorting. Similarly, in the engaged sample, 58% of ribosomes (189,099 particles) were unratcheted and contained tRNA, while 28% of these particles (52,061) were bound by SRP. In order to isolate the population of ribosomes bound to both SRP and eEF2 in the engaged sample, a mask for SRP was used to further sub-classify the population of ribosomes bound to eEF2 (20%: 64,416) resulting in a final population of 8418 particles.

Final 3D refinements of the resulting populations were performed without external masking, utilizing statistical movie processing (*Bai et al., 2013*), and particle polishing (*Scheres, 2014*). This resulted in final reconstructions at overall resolution of 3.9 Å, 3.75 Å, and 5.0 Å for the scanning, engaged, and ternary complex structures, respectively, using the gold-standard FSC = 0.143 criteria (*Scheres and Chen, 2012*).

## Model building and refinement

All models were initially built and refined in the density for the higher resolution engaged complex, and then rigid body fit into the density for the scanning sample. The resulting scanning model required only minor adjustments to the M domain to fit the respective density (e.g., *Figure 6*). The 60S subunit, 40S body, and 40S head were individually placed using models of the porcine ribosome (*Voorhees et al., 2014*), and the P-site tRNA was homology modelled using the bacterial structure (*Voorhees et al., 2009*). SRP was built using a combination of crystal structures of its individual domains, which were modified to fit the observed density as described below.

Within the S domain, the N-domain of SRP54 was built primarily using the solution structure of the human domain (PDB ID: 1WGW) with minor modification to the loops that interact with the ribosome. The G-domain was modelled using the structure from (*Janda et al., 2010*) (PDB ID: 3KL4). The

M-domain itself was modelled using the crystal structure of the human M-domain (*Clemons et al., 1999*) (PDB ID: 1QB2), as well as models of the occupied M-domain from homologous species (*Janda et al., 2010*; *Hainzl et al., 2011*), with modifications to account for the interface with the ribosome and other SRP components. Due to lower resolution in this region, the S-domain RNA, SRP19, and a portion of SRP68 were simply rigid-body fit using the model from *Grotwinkel et al. (2014)* (PDB ID: 4P3E).

The Alu domain was built based on models from (*Weichenrieder et al., 2000*), including the proteins SRP9 and SRP14 (PDB ID: 1E8O), and the Alu RNA (assembled from PDB IDs: 1E8O and 1E8S). All models were built in COOT (*Emsley et al., 2010*). Refinement of the 40S subunit plus Alu domain, and 60S subunit plus SRP54, were carried out individually using using REFMAC v5.8 (*Murshudov et al., 2011*) as previously described (*Amunts et al., 2014*; *Brown et al., 2015*). Secondary structure restraints were generated in ProSMART (*Nicholls et al., 2012*), and nucleic acid base-pairing and stacking restraints were generated as before (*Amunts et al., 2014*) and were maintained throughout refinement to prevent over-fitting. Local resolution was calculated using ResMap (*Kucukelbir et al., 2014*) and all figures were generated using Pymol (*DeLano, 2006*) and Chimera (*Goddard et al., 2007*).

## Acknowledgements

We thank Felix de Haas, Vinothkumar Ragunath, and Christos Savva for help with data collection; Susan Shao for help with initial sample preparation; Tim Stevens for bioinformatics analysis; Shaoxia Chen, Greg McMullan, Jake Grimmett and Toby Darling for technical support; and Garib Murshudov for help with model building and refinement. This work was supported by the UK Medical Research Council (MC_UP_A022_1007 to RSH) and a Wellcome Trust postdoctoral fellowship (RMV). Cryo-EM density maps have been deposited with the EMDataBank with the following accession codes: EMDB-3037 (engaged SRP-RNC complex), EMDB-3045 (scanning SRP-RNC complex), and EMDB-3046 (SRP-RNC ternary complex with eEF2). The Protein Data Bank accession numbers for the scanning and engaged complexes are 3JAN and 3JAJ, respectively.

## Additional information

### Competing interests

RSH: Reviewing editor, *eLife.* The other author declares that no competing interests exist.

### Funding

| Funder | Grant reference | Author |
|---|---|---|
| Medical Research Council (MRC) | MC_UP_A022_1007 | Ramanujan S Hegde |
| Wellcome Trust | Postdoctoral Fellowship | Rebecca M Voorhees |

The funders had no role in study design, data collection and interpretation, or the decision to submit the work for publication.

### Author contributions

RMV, Conception and design, Acquisition of data, Analysis and interpretation of data, Drafting or revising the article; RSH, Conception and design, Analysis and interpretation of data, Drafting or revising the article

## Additional files

### Major datasets

The following previously published datasets were used:

| Author(s) | Year | Dataset title | Dataset ID and/or URL | Database, license, and accessibility information |
|---|---|---|---|---|
| Li H, Tomizawa T, Koshiba S, Inoue M, Kigawa T, Yokoyama S | 2004 | Solution Structure of the N-terminal Domain of Mouse Putative Signal Recognition Particle 54 (SRP54) | http://www.rcsb.org/pdb/explore/explore.do?structureId=1WGW | Publicly available at RCSB Protein Data Bank (Accession No. 1WGW). |

| Author(s) | Year | Dataset title | Dataset ID and/or URL | Database, license, and accessibility information |
|---|---|---|---|---|
| Janda CY, Li J, Oubridge C, Hernandez H, Robinson CV, Nagai K | 2010 | Recognition of a signal peptide by the signal recognition particle | http://www.rcsb.org/pdb/explore/explore.do?structureId=3KL4 | Publicly available at RCSB Protein Data Bank (Accession No. 3KL4). |
| Hainzl T, Huang S, Merilainen G, Brannstrom K, Sauer-Eriksson AE | 2011 | Crystal structure of a signal sequence bound to the signal recognition particle | http://www.rcsb.org/pdb/explore/explore.do?structureId=3NDB | Publicly available at RCSB Protein Data Bank (Accession No. 3NDB). |
| Clemons WM Jr, Gowda K, Black SD, Zwieb C, Ramakrishnan V | 1999 | Crystal structure of the conserved subdomain of human protein SRP54m at 2.1A resolution: Evidence for the mechanism of signal peptide binding | http://www.rcsb.org/pdb/explore/explore.do?structureId=1QB2 | Publicly available at RCSB Protein Data Bank (Accession No. 1QB2). |
| Grotwinkel JT, Wild K, Segnitz B, Sinning I | 2014 | Structure of the human SRP S domain | http://www.rcsb.org/pdb/explore/explore.do?structureId=4P3E | Publicly available at RCSB Protein Data Bank (Accession No. 4P3E). |
| Weichenrieder O, Wild K, Strub K, Cusack S | 2000 | Core of the Alu domain of the mammalian SRP | http://www.rcsb.org/pdb/explore/explore.do?structureId=1E8O | Publicly available at RCSB Protein Data Bank (Accession No. 1E8O). |
| Weichenrieder O, Wild K, Strub K, Cusack S | 2000 | Alu domain of the mammalian SRP (potential Alu retroposition intermediate) | http://www.rcsb.org/pdb/explore/explore.do?structureId=1E8S | Publicly available at RCSB Protein Data Bank (Accession No. 1E8S). |

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
