## [Decision Letter]

Thank you for sending your work entitled “Structures of the scanning and engaged states of the mammalian SRP-ribosome complex” for consideration at *eLife*. Your article has been favorably evaluated by John Kuriyan (Senior editor and Reviewing editor) and three reviewers, one of whom, Peter Walter, has agreed to reveal his identity.

As you will see, all three reviewers are in agreement that your work represents a major advance. The comments of the reviewers are such that you should be able to deal with them in a straightforward way. Because this is the case we have not merged the three reviews, and instead have simply collated them below this letter. Please note that the reviewers all ask for a better discussion of the electron microscopy results, and your revised manuscript should carefully take this point into consideration.

Review 1:

Here, the authors determined the structures of SRP-RNC complexes in two states using cryo-EM. Using a rabbit reticulocyte lysate in vitro translation system and truncated mRNAs, stalled RNC complexes were generated that contain nascent polypeptide chains with a transmembrane domain (TMD) that was either (1) exposed from the ribosomal exit tunnel (hereafter “engaged”), or (2) buried in the exit tunnel (hereafter “scanning”). These RNCs were purified via the nascent chain's 3xFLAG N-terminus. The authors found that SRP co-purified with both these complexes, but was absent (or depleted) from RNCs containing three arginine point mutations in the TMD, thus demonstrating SRP's specificity for RNCs containing the hydrophobic TMD.

From here, the structures of the scanning SRP-RNC and engaged SRP-RNC were solved to ∼3.7-3.8 Å overall resolution, though the resolution of SRP is lower. SRP's lower local resolution limited interpretation of its RNC-bound conformation to just its Alu domain and SRP54 protein. Overall, the two structures are very comparable, with the main difference being the electron density of the SRP M-domain's peptide binding groove. The authors fit the electron density with the available crystal structures of SRP54 and assign the extra density to two alpha helices (αC1 and αC2) absent in the crystal structures of truncated SRP54. The density of these helices differs between the engaged and scanning states, and the authors interpret these changes in density to movement of the helices to either occupy the peptide-binding groove in the absence of a nascent chain's TMD, or act as a lid for the groove in the presence of a TMD.

Since SRP's Alu domain occupies the GTPase binding site on the ribosome, the authors queried whether it competes with GTPase translation factors for binding to the RNC. They chose to test this hypothesis using the GTPase Hbs1. By taking an immunoprecipitation approach, the authors enriched for Hbs1-bound RNCs and found that a subset of these was co-occupied by SRP, indicating that these three molecules can exist as a ternary complex (RNC-SRP-Hbs1). Since SRP was found stably bound (over the time course of their experiments) to RNCs in both its scanning and engaged state, the authors speculate that unlike the smaller, prokaryotic SRP, mammalian SRP may remain bound to RNCs by its S domain throughout RNC translation, while its Alu domain may repeatedly bind and dissociate from the ribosome's GTPase binding site by virtue of the flexible linker connecting the S and Alu domains. By remaining stably bound to the RNC by its S domain, SRP could continuously survey nascent polypeptide chains for its hydrophobic substrates, and prevent their exposure to the (aqueous) environment.

The structures solved in this study are the first SRP-RNC complexes determined from homologous mammalian components. The high resolution structure of the complex suggest a new model where the two C-terminal alpha helices in SRP's M domain act as a lid for the peptide-binding groove in the engaged state, whereas they occupy the groove in the absence of a peptide during scanning. Additionally, based on biochemical data and the structures, the authors propose a model for SRP engagement in eukaryotes. They propose that SRP remains stably bound to RNC by its S domain, while its Alu domain may dissociate and re-bind during nascent chain scanning, where the mechanism differs distinctly from the well-studied prokaryotic system. Overall, this is an exciting advance that is appropriate for publication in *eLife*.

Specific comments:

1) Figure 2, and discussion of weak density in the scanning complex: It is unclear why it could be “possible that the nascent chain is present at lower occupancy” considering that the RNCs are purified by the 3x-FLAG tag on the nascent chain. This needs to be clarified.

2) The story would have been strengthened by the use of a translation elongation factor in place of Hbs1. This seems a major omission and the authors should explain why this wasn't done.

3) In Table 1, additional information about the classification of particles is needed. What was the total number of selected particles in the first pass? How many contained just a P-site tRNA? How many contained just SRP? How many had both a P-site tRNA and SRP?

4) If SRP competes with translation elongation factors for binding to the ribosomal GTPase center, yet remains bound to the RNC via its S domain, why isn't the scanning SRP-RNC complex more compositionally heterogeneous? I.e. shouldn't there be a subset of the particles that contain the ternary complex of SRP-RNC-eEF2? This should be discussed/clarified.

Review 2:

In the manuscript by Voorhees and Hegde, the authors present an elegant model for how the Signal Recognition Particle (SRP) scans ribosome nascent chains (RNCs) to engage the first hydrophobic trans-membrane domain (TMD) that emerges from the ribosome, for subsequent targeting to the endoplasmic reticulum. They use a combination of cryo-EM reconstructions and biochemistry to identify the features of a “scanning” SRP complex with the ribosome, in which the TMD is still in the exit tunnel, and the “engaged” SRP complex, in which the TMD is fully exposed from the ribosome. From these experiments, the authors propose a compelling biophysical model for how SRP is autoinhibited by a C-terminal amphipathic helix that allows dynamic sampling of emerging nascent chains. If a sufficiently hydrophobic stretch emerges from the ribosome, it could compete for binding to the hydrophobic cleft in the SRP M-domain to form the “engaged” complex. In this state the TMD is then protected by the amphipathic helix from exposure to the cytosol until the RNC is properly targeted to the ER.

Overall, this an excellent paper that should be published with only minor changes. It establishes a framework for SRP function that will certainly stimulate many interesting follow-up studies.

Specific comments:

1) The authors should be a little clearer about the topology of alpha-C2 relative to the M-domain. For example, would the corresponding C-terminal helix in bacterial Ffh be able to bind to the M domain, in terms of the linker? Also, is the C-terminal helix conserved in other bacteria, in particular *B. subtilis*, which has a more eukaryotic-like SRP?

2) In the Discussion section, the authors argue that based on their scanning complex structure, SRP appears to favor the unratcheted state. But this is somewhat circular reasoning. The 3D classification eliminated a very large number of particles prior to the final refinement (27,415 particles out of 639,184). I think the authors should heavily qualify these statements, and instead rely more on published results presented in the same section.

3) The end of the sixth paragraph of the Discussion section is a bit confusing. I'm not quite sure what point the authors are making. Further, the authors should use species-consistent naming conventions for the proteins listed here (all mammalian with yeast in parentheses, for example).

4) Figure 2—figure supplement 2 panels need to be a little higher resolution in terms of dpi.

5) It's not quite clear from Table 1 how the model was refined. Was it refined as 60S + S domain separately from 40S + Alu domain? Or globally?

Review 3:

The paper by Voorhees and Hegde describes biochemical and EM characterization of mammalian ribosomes bound to the SRP in a 'scanning' and 'engaged' state. The work is well done and presented clearly. This provides the first detailed picture of the mammalian ribosome/SRP interactions. Novelty in the mechanism results from the demonstration that the M-domain contains two additional helices that block the hydrophobic groove and stabilize TM interactions and the fact that there are no structural differences in the ribosome that can clearly distinguish the scanning and egaged binding modes. This last point is quite exciting as it allows them to posit a striking hypothesis that the role of the Alu domain is to sense the state translation state of the ribosome, which may be sensitive to hydrophobic stretches in the nascent chain. This is really interesting and for a general point I found it unsatisfying that the discussion of differences in the ribosome of the two structures was not brought up until the Discussion. I found myself wanting to know this throughout reading the manuscript and it could have at least been brought up earlier. I think the work is clearly at the level strived for by *eLife* and would only make suggestions about clarification as highlighted below.

Specific comments:

Abstract: I don't understand the use of 'auto-inhibited' as the story is presented. Certainly the hydrophobic binding groove is occluded but this seems to be for protection of the hydrophobic surfaces not inhibition.

Introduction, fourth paragraph: I'm not sure how these references relate to the story being told here. Pellecchia et al. (assuming I have the correct reference as there is no citation in the references) describes the isolated structure of the isolated β-domain of DnaK which seems inconsistent with more recent structures of the open ATP-bound form of Hsp70. The second reference (to the corresponding authors work also not in the references) also seems like a poor comparison as the Get3 structures in that paper are supposedly bound to substrate and, therefore, engaged. Perhaps the reference was meant to an early open Get3 structure? I would just delete the sentence.

In the subsection “Biochemical analysis of SRP complexes in scanning and engaged states”, the expression 'similar levels to the ribosome' seems pretty strong. Looking at the gel it seems like there is not a stoichiometric pull-down. Clearly the ribosome-nascent chain as shown subsequently is capturing the bulk of the SRP, however, it isn't stoichiometric and I wouldn't expect it to be as there is likely to be more translating ribosomes than SRP. I suppose this just requires re-wording.

In the subsection headed “Cryo-EM structures of SRP in its scanning and engaged states”, I found the overall description of the resolution to be a bit disingenuous. Presumably, this resolution is largely driven by the ribosome and most of the SRP density is at much lower resolution. This should be better expressed in the text as it definitely affects the interpretation, especially in regions such as the M-domain where there are no atomic level details.

In the second paragraph of the subsection headed “The SRP54 M-domain before and after substrate engagement”: The two helices have been matched to sequence in Figure 5—figure supplement 1. It would be useful to discuss the logic of this. My guess is that αC2 is more hydrophobic making it better fit the groove but that should be explained.

Discussion, fourth paragraph, last sentence: This sentence again references a paper that doesn't directly demonstrate 'shielding'. These regions in the Get3-Tail anchor substrate complex that 'shield' are unexpectedly disordered if they function similarly to the SRP C-terminus.

Figure 1: Could these numbers be expressed as they are in the text (in terms of TM length)? I found it hard to correlate while reading.

Figure 4: It might be nice to add a supplemental figure highlighting the uL11/uL10 interface as it is discussed in the text.

Figure 5: It would be useful to see how the additional two-helices fit into the architecture of the M-domain. Perhaps just a supplemental showing that structure and how it would connect to the rest of the known structure.

Figure 7: It isn't clear in the 'scanning' that you get Hbs1 dependent loss of SRP binding as the difference is hard to detect between concentrations. Perhaps just adding quantification here would help.

---

## [Author Response]

Review 1:

*1)*
Figure 2*, and discussion of weak density in the scanning complex: It is unclear why it could be* “*possible that the nascent chain is present at lower occupancy*” *considering that the RNCs are purified by the 3x-FLAG tag on the nascent chain. This needs to be clarified*.

Though all isolated RNCs presumably contain a nascent chain upon purification, subsequent processing for EM analysis, including application to grids and freezing, results in some hydrolysis of the peptidyl-tRNA and release of the nascent chain. This is consistent with the observation that only 27 or 58% of ribosome particles contain a P-site tRNA in the scanning and engaged populations, respectively. We therefore cannot know whether the weaker density for the nascent chain in the scanning reconstruction is due to decreased occupancy or increased flexibility. We have added a sentence to the Methods section to clarify this point (please see the subsection headed “Image processing”).

*2) The story would have been strengthened by the use of a translation elongation factor in place of Hbs1. This seems a major omission and the authors should explain why this wasn't done*.

We apologise for not clarifying this point in the original manuscript. Dominant negative Hbs1 (DN-Hbs1) was utilized because it preferentially binds unratcheted ribosomes lacking an mRNA codon in the A site, the state represented by our stalled SRP-ribosome-nascent chain complexes. Association of eEF1 requires specific recognition of the A-site codon, while dominant negative eEF2 would presumably bind most stably to a ratcheted ribosome. Furthermore, addition of dominant negative versions of either eEF1 or eEF2 would interfere with translation elongation. These technical obstacles necessitated the use of Hbs1.This logic is briefly described when we first introduce the use of Hbs1 (subsection “Analysis of SRP-ribosome interactions during translation elongation”).

*3) In*
Table 1*, additional information about the classification of particles is needed. What was the total number of selected particles in the first pass? How many contained just a P-site tRNA? How many contained just SRP? How many had both a P-site tRNA and SRP?*

We have added a schematic detailing the computational sorting of particles that addresses these questions (Figure 2—figure supplement 1).

*4) If SRP competes with translation elongation factors for binding to the ribosomal GTPase center, yet remains bound to the RNC via its S domain, why isn't the scanning SRP-RNC complex more compositionally heterogeneous? I.e. shouldn't there be a subset of the particles that contain the ternary complex of SRP-RNC-eEF2? This should be discussed/clarified*.

This is an astute point that prompted us to go back and specifically look in our dataset for this subset of particles. We find that, as predicted by the referee, ∼13% of particles with eEF2 in the engaged sample also contain the S domain of SRP. At low threshold, most of the remainder of SRP can also be observed and has clearly moved away from the GTPase centre, where eEF2 is bound. This is a compelling additional argument for our proposed model, which we have now included as Figure 7 and accompanying figure supplement.

It is worth noting that while the scanning sample also contains ∼40% of particles bound to eEF2, no density was observed surrounding the exit tunnel. This may be due in part to the observation that association of SRP to the ribosome in the scanning mode is less stable than in the engaged mode (i.e. Figure 7), reducing its recovery during purification.

Review 2:

*1) The authors should be a little clearer about the topology of alpha-C2 relative to the M-domain. For example, would the corresponding C-terminal helix in bacterial Ffh be able to bind to the M domain, in terms of the linker? Also, is the C-terminal helix conserved in other bacteria, in particular* B. subtilis*, which has a more eukaryotic-like SRP?*

Given the resolution in this region, we cannot unambiguously determine the topology of either of the two C-terminal helices of the M domain. In order to help to better orient the reader we have indicated the C-terminus of the M domain that would connect to these helices with an asterisk in the relevant figures (Figures 5 and 6).

The other point about bacterial Ffh and its conservation is interesting, but we realised this is all rather speculative given our exclusive focus on the eukaryotic system. We have therefore decided to omit the speculation about the bacterial system until a more thorough bioinformatics and structural analysis can be performed. Of note, the presence of a C-terminal methionine rich region does appear to be conserved across bacteria, but helicity predictions appear to be more variable. In *T. aquaticus* for example, this region is predicted to be helical and amphipathic, but is less clear in *B. subtilis*. It is possible that the role of this helix, and therefore the evolutionary pressure to conserve its amphipathic nature, is more important in higher organisms in part due to their more complex cytosolic environment and low tolerance for aggregation.

*2) In the Discussion section, the authors argue that based on their scanning complex structure, SRP appears to favor the unratcheted state. But this is somewhat circular reasoning. The 3D classification eliminated a very large number of particles prior to the final refinement (27,415 particles out of 639,184). I think the authors should heavily qualify these statements, and instead rely more on published results presented in the same section*.

We agree with the reviewer and have now modified the text accordingly.

*3) The end of the sixth paragraph of the Discussion section is a bit confusing. I'm not quite sure what point the authors are making. Further, the authors should use species-consistent naming conventions for the proteins listed here (all mammalian with yeast in parentheses, for example)*.

We have re-worded this section to hopefully clarify our point.

*4)*
Figure 2—figure supplement 2
*panels need to be a little higher resolution in terms of dpi.*

All images were downsampled to simplify the review process, but will be provided at high resolution for publication.

*5) It's not quite clear from*
Table 1
*how the model was refined. Was it refined as 60S + S domain separately from 40S + Alu domain? Or globally?*

This has been clarified in the Methods section (in the subsection headed “Model building and refinement”).

Review 3:

*The paper by Voorhees and Hegde describes biochemical and EM characterization of mammalian ribosomes bound to the SRP in a 'scanning' and 'engaged' state. The work is well done and presented clearly. This provides the first detailed picture of the mammalian ribosome/SRP interactions. Novelty in the mechanism results from the demonstration that the M-domain contains two additional helices that block the hydrophobic groove and stabilize TM interactions and the fact that there are no structural differences in the ribosome that can clearly distinguish the scanning and egaged binding modes. This last point is quite exciting as it allows them to posit a striking hypothesis that the role of the Alu domain is to sense the state translation state of the ribosome, which may be sensitive to hydrophobic stretches in the nascent chain. This is really interesting and for a general point I found it unsatisfying that the discussion of differences in the ribosome of the two structures was not brought up until the Discussion. I found myself wanting to know this throughout reading the manuscript and it could have at least been brought up earlier. I think the work is clearly at the level strived for by* eLife *and would only make suggestions about clarification as highlighted below*.

We apologise for this omission. We now specifically point out at the outset the absence of an observable conformational change in the ribosome when a TMD is inside the tunnel (please see the subsection “Cryo-EM structures of SRP in its scanning and engaged states”). Of course, we reserve our detailed discussion of this point, including our mechanistic model for tunnel-mediated recruitment, until the Discussion section.

*Specific comments*:

*Abstract: I don't understand the use of 'auto-inhibited' as the story is presented. Certainly the hydrophobic binding groove is occluded but this seems to be for protection of the hydrophobic surfaces not inhibition*.

We believe that groove occlusion serves a dual function: to protect the hydrophobic groove, and inhibit promiscuous binding. It is this latter function that led to our use of this term.

*Introduction, fourth paragraph: I'm not sure how these references relate to the story being told here. Pellecchia et al. (assuming I have the correct reference as there is no citation in the references) describes the isolated structure of the isolated β-domain of DnaK which seems inconsistent with more recent structures of the open ATP-bound form of Hsp70. The second reference (to the corresponding authors work also not in the references) also seems like a poor comparison as the Get3 structures in that paper are supposedly bound to substrate and, therefore, engaged. Perhaps the reference was meant to an early open Get3 structure? I would just delete the sentence*.

We apologise for not including the references. The point was that occlusion of the substrate binding groove of hydrophobic proteins is likely a general phenomenon across biology. The reference to the earlier Get3 structure has now been included.

In the subsection “Biochemical analysis of SRP complexes in scanning and engaged states”, the expression 'similar levels to the ribosome' seems pretty strong. Looking at the gel it seems like there is not a stoichiometric pull-down. Clearly the ribosome-nascent chain as shown subsequently is capturing the bulk of the SRP, however, it isn't stoichiometric and I wouldn't expect it to be as there is likely to be more translating ribosomes than SRP. I suppose this just requires re-wording.

The text has been altered as suggested (Results).

*In the subsection headed “Cryo-EM structures of SRP in its scanning and engaged states”, I found the overall description of the resolution to be a bit disingenuous. Presumably, this resolution is largely driven by the ribosome and most of the SRP density is at much lower resolution. This should be better expressed in the text as it definitely affects the interpretation, especially in regions such as the M-domain where there are no atomic level details*.

As we show density maps for all regions of SRP that are discussed in detail, as well as the local resolution plots, we did not feel our discussion of the resolution was disingenuous. Nevertheless, we now explicitly state the range of local resolutions for SRP in the text in the sentence immediately following the statement of overall resolutions (subsection headed “Cryo-EM structures of SRP in its scanning and engaged states”).

*In the second paragraph of the subsection headed “The SRP54 M-domain before and after substrate engagement”: The two helices have been matched to sequence in*
Figure 5—figure supplement 1*. It would be useful to discuss the logic of this. My guess is that αC2 is more hydrophobic making it better fit the groove but that should be explained*.

We do not have sufficient resolution to confidently assign the two regions of helical density to specific sequence, but have provisionally assigned them based on their apparent length in comparison to secondary structure predictions. This has been clarified (please see “The SRP54 M-domain before and after substrate engagement”).

*Discussion, fourth paragraph, last sentence: This sentence again references a paper that doesn't directly demonstrate 'shielding'. These regions in the Get3-Tail anchor substrate complex that 'shield' are unexpectedly disordered if they function similarly to the SRP C-terminus*.

The structure of the substrate bound Get3 does suggest that the substrate is at least partially shielded, despite the fact that these ‘lid’ helices are somewhat poorly ordered. Note that this conclusion was not based solely on structural considerations, but also supported by photo-crosslinking experiments in that paper.

Figure 1*: Could these numbers be expressed as they are in the text (in terms of TM length)? I found it hard to correlate while reading*.

We have ed to leave this unchanged to avoid unnecessary clutter in an already busy figure. Note that a diagram showing the residue numbers of the TMD is shown in panel A, permitting a reader to correlate truncation point with TMD length.

Figure 4: *It might be nice to add a supplemental figure highlighting the uL11/uL10 interface as it is discussed in the text.*

Unfortunately, as stated in the text, uL10/uL11 are not sufficiently well ordered to confidently interpret the molecular details of the interaction surface between the ribosomal stalk and the Alu domain.

Figure 5*: It would be useful to see how the additional two-helices fit into the architecture of the M-domain. Perhaps just a supplemental showing that structure and how it would connect to the rest of the known structure*.

Given the resolution in this region, we cannot unambiguously determine the topology of either of the two C-terminal helices of the M domain. In order to help to better orient the reader we have indicated the C-terminus of the M domain with an asterisk in the appropriate figures (Figures 5 and 6).

Figure 7*: It isn't clear in the 'scanning' that you get Hbs1 dependent loss of SRP binding as the difference is hard to detect between concentrations. Perhaps just adding quantification here would help*.

We now note in the text that ∼70% is lost at the highest concentration as estimated by densitometry (please see “Analysis of SRP-ribosome interactions during translation elongation”).